# ADVERSARIAL TRAINING SHOULD BE CAST AS A NON-ZERO-SUM GAME

**Alexander Robey**[*]
University of Pennsylvania
arobey1@upenn.edu

**Fabian Latorre**[*]
LIONS, EPFL
fabian.latorre@epfl.ch

**George J. Pappas**
University of Pennsylvania
pappasg@upenn.edu

**Hamed Hassani**
University of Pennsylvania
hassani@upenn.edu

**Volkan Cevher**
LIONS, EPFL
volkan.cevher@epfl.ch

## ABSTRACT

One prominent approach toward resolving the adversarial vulnerability of deep neural networks is the two-player zero-sum paradigm of adversarial training, in which predictors are trained against adversarially chosen perturbations of data. Despite the promise of this approach, algorithms based on this paradigm have not engendered sufficient levels of robustness and suffer from pathological behavior like robust overfitting. To understand this shortcoming, we first show that the commonly used surrogate-based relaxation used in adversarial training algorithms voids all guarantees on the robustness of trained classifiers. The identification of this pitfall informs a novel non-zero-sum bilevel formulation of adversarial training, wherein each player optimizes a different objective function. Our formulation yields a simple algorithmic framework that matches and in some cases outperforms state-of-the-art attacks, attains comparable levels of robustness to standard adversarial training algorithms, and does not suffer from robust overfitting.

## 1 INTRODUCTION

A longstanding disappointment in the machine learning (ML) community is that deep neural networks (DNNs) remain vulnerable to seemingly innocuous changes to their input data, including nuisances in visual data (Laidlaw et al., 2020; Hendrycks & Dietterich, 2019), sub-populations (Santurkar et al., 2021; Koh et al., 2021), and distribution shifts (Xiao et al., 2021; Arjovsky et al., 2019; Robey et al., 2021). Prominent amongst these vulnerabilities is the setting of *adversarial examples*, wherein it has been conclusively shown that imperceptible, adversarially-chosen perturbations can fool state-of-the-art classifiers parameterized by DNNs (Szegedy et al., 2013; Biggio et al., 2013). In response, a plethora of research has proposed so-called adversarial training (AT) algorithms (Madry et al., 2018; Goodfellow et al., 2015), which are designed to improve robustness against adversarial examples.

AT is ubiquitously formulated as a *two-player zero-sum* game, where both players—often referred to as the *defender* and the *adversary*—respectively seek to minimize and maximize the classification error. However, this zero-sum game is not implementable in practice as the discontinuous nature of the classification error is not compatible with first-order optimization algorithms. To bridge this gap between theory and practice, it is commonplace to replace the classification error with a smooth surrogate loss (e.g., the cross-entropy loss) which is amenable to gradient-based optimization (Madry et al., 2018; Zhang et al., 2019). And while this seemingly harmless modification has a decades-long tradition in the ML literature due to the guarantees it imparts on non-adversarial objectives (Bartlett et al., 2006; Shalev-Shwartz & Ben-David, 2014; Roux, 2017), there is a pronounced gap in the literature regarding the implications of this relaxation on the standard formulation of AT.

As the field of robust ML has matured, surrogate-based AT algorithms have collectively resulted in steady progress toward stronger attacks and robust defenses (Croce et al., 2020a). However, despite these advances, recent years have witnessed a plateau in robustness measures on popular leaderboards, resulting in the widely held beliefs that robustness and accuracy may be irreconcilable (Tsipras et al.,

---

[*]The first two authors contributed equally.

2019; Dobriban et al., 2020) and that robust generalization requires significantly more data (Schmidt et al., 2018; Chen et al., 2020). Moreover, various phenomena such as robust overfitting (Rice et al., 2020) have indicated that progress has been overestimated (Croce & Hein, 2020). To combat these pitfalls, state-of-the-art algorithms increasingly rely on ad-hoc regularization schemes (Kannan et al., 2018; Chan et al., 2020), weight perturbations (Wu et al., 2020; Sun et al., 2021), and heuristics such as multiple restarts, carefully crafted learning rate schedules, and convoluted stopping conditions, all of which contribute to an unclear set of best practices and a growing literature concerned with identifying flaws in various AT schemes (Latorre et al., 2023).

Motivated by these challenges, we argue that the pervasive surrogate-based zero-sum approach to AT suffers from a fundamental flaw. Our analysis of the standard minimax formulation of AT reveals that maximizing a surrogate like the cross-entropy provides no guarantee that the the classification error will increase, resulting in weak adversaries and ineffective AT algorithms. In identifying this shortcoming, we prove that to preserve guarantees on the optimality of the classification error objective, the defender and the adversary must optimize different objectives, resulting in a *non-zero-sum* game. This leads to a novel, yet natural *bilevel* formulation (Bard, 2013) of AT in which the defender minimizes an upper bound on the classification error, while the attacker maximizes a continuous reformulation of the classification error. We then propose an algorithm based on our formulation which is free from heuristics and ad hoc optimization techniques. Our empirical evaluations reveal that our approach matches the test robustness achieved by the state-of-the-art, yet highly heuristic approaches such as AutoAttack, and that it eliminates robust overfitting.

**Contributions.** Our contributions are as follows.

- **New formulation for adversarial robustness.** Starting from the discontinuous minmax formulation of AT with respect to the 0-1 loss, we derive a novel continuous bilevel optimization formulation, the solution of which *guarantees* improved robustness against the optimal adversary.
- **New adversarial training algorithm.** We derive BETA, a new, heuristic-free algorithm based on our bilevel formulation which offers competitive empirical robustness on CIFAR-10.
- **Elimination of robust overfitting.** Our algorithm does not suffer from robust overfitting. This suggest that robust overfitting is an artifact of the use of improper surrogates in the original AT paradigm, and that the use of a correct optimization formulation is enough to solve it.
- **State-of-the-art robustness evaluation.** We show that our proposed optimization objective for the adversary yields a simple algorithm that matches the performance of the state-of-the-art, yet highly complex AutoAttack method, on state-of-the-art robust classifiers trained on CIFAR-10.

## 2 THE PROMISES AND PITFALLS OF ADVERSARIAL TRAINING

### 2.1 PRELIMINARIES: TRAINING DNNs WITH SURROGATE LOSSES

We consider a $K$-way classification setting, wherein data arrives in the form of instance-label pairs $(X, Y)$ drawn i.i.d. from an unknown joint distribution $\mathcal{D}$ taking support over $\mathcal{X} \times \mathcal{Y} \subseteq \mathbb{R}^d \times [K]$, where $[K] := \{1, \ldots, K\}$. Given a suitable hypothesis class $\mathcal{F}$, one fundamental goal in this setting is to select an element $f \in \mathcal{F}$ which correctly predicts the label $Y$ of a corresponding instance $X$. In practice, this hypothesis class $\mathcal{F}$ often comprises functions $f_\theta : \mathbb{R}^d \to \mathbb{R}^K$ which are parameterized by a vector $\theta \in \Theta \subset \mathbb{R}^p$, as is the case when training DNNs. In this scenario, the problem of learning a classifier that correctly predicts $Y$ from $X$ can written as follows:

$$\min_{\theta \in \Theta} \mathbb{E} \left\{ \arg\max_{i \in [K]} f_\theta(X)_i \neq Y \right\} \tag{1}$$

Here $f_\theta(X)_i$ denotes the $i^{\text{th}}$ component of the logits vector $f_\theta(X) \in \mathbb{R}^K$ and we use the notation $\{A\}$ to denote the indicator function of an event $A$, i.e., $\{A\} := \mathbb{I}_A(\cdot)$. In this sense, $\{\arg\max_{i \in [K]} f_\theta(X)_i \neq Y\}$ denotes the *classification error* of $f_\theta$ on the pair $(X, Y)$.

Among the barriers to solving (1) in practice is the fact that the classification error is a discontinuous function of $\theta$, which in turn renders continuous first-order methods intractable. Fortunately, this pitfall can be resolved by minimizing a surrogate loss function $\ell : [k] \times [k] \to \mathbb{R}$ in place of the classification error (Shalev-Shwartz & Ben-David, 2014, §12.3). For minimization problems, surrogate losses are chosen to be differentiable *upper bounds* of the classification error of $f_\theta$ in the sense that

$$\left\{ \arg\max_{i \in [K]} f_\theta(X)_i \neq Y \right\} \leq \ell(f_\theta(X), Y). \tag{2}$$

This inequality gives rise to a differentiable counterpart of (1) which is amenable to minimization via first-order methods and can be compactly expressed in the following optimization problem:

$$\min_{\theta \in \Theta} \mathbb{E}\, \ell(f_\theta(X), Y). \tag{3}$$

Examples of commonly used surrogates are the hinge loss and the cross-entropy loss. Crucially, the inequality in (2) guarantees that the problem in (3) provides a solution that decreases the classification error (Bartlett et al., 2006), which, as discussed above, is the primary goal in supervised classification.

## 2.2 THE PERVASIVE SETTING OF ADVERSARIAL EXAMPLES

For common hypothesis classes, it is well-known that classifiers obtained by solving (3) are sensitive to adversarial examples (Szegedy et al., 2013; Biggio et al., 2013), i.e., given an instance label pair $(X, Y)$, it is relatively straightforward to find perturbations $\eta \in \mathbb{R}^d$ with small norm $||\eta|| \leq \epsilon$ for some fixed $\epsilon > 0$ such that

$$\arg\max_{i \in [K]} f_\theta(X)_i = Y \qquad \text{and} \qquad \arg\max_{i \in [K]} f_\theta(X + \eta)_i \neq \arg\max_{i \in [K]} f_\theta(X)_i. \tag{4}$$

The task of finding such perturbations $\eta$ which cause the classifier $f_\theta$ to misclassify perturbed data points $X + \eta$ can be compactly cast as the following maximization problem:

$$\eta^\star \in \arg\max_{\eta : ||\eta|| \leq \epsilon} \left\{ \arg\max_{i \in [K]} f_\theta(X + \eta)_i \neq Y \right\} \tag{5}$$

Here, if both of the expressions in (4) hold for the perturbation $\eta = \eta^\star$, then the perturbed instance $X + \eta^\star$ is called an *adversarial example* for $f_\theta$ with respect to the instance-label pair $(X, Y)$.

Due to prevalence of adversarial examples, there has been pronounced interest in solving the robust analog of (1), which is designed to find classifiers that are insensitive to small perturbations. This robust analog is ubiquitously written as the following a two-player zero-sum game with respect to the discontinuous classification error:

$$\min_{\theta \in \Theta} \mathbb{E} \left[ \max_{\eta : ||\eta|| \leq \epsilon} \left\{ \arg\max_{i \in [K]} f_\theta(X + \eta)_i \neq Y \right\} \right] \tag{6}$$

An optimal solution $\theta^\star$ for (6) yields a model $f_{\theta^\star}$ that achieves the lowest possible classification error despite the presence of adversarial perturbations. For this reason, this problem—wherein the interplay between the maximization over $\eta$ and the minimization over $\theta$ comprises a two-player zero-sum game— is the starting point for numerous algorithms which aim to improve robustness.

## 2.3 SURROGATE-BASED APPROACHES TO ROBUSTNESS

As discussed in § 2.1, the discontinuity of the classification error complicates the task of finding adversarial examples, as in (5), and of training against these perturbed instances, as in (6). One appealing approach toward overcoming this pitfall is to simply deploy a surrogate loss in place of the classification error inside (6), which gives rise to the following pair of optimization problems:

$$\eta^\star \in \arg\max_{\eta : ||\eta|| \leq \epsilon} \ell(f_\theta(X + \eta), Y) \quad (7) \qquad \min_{\theta \in \Theta} \mathbb{E} \left[ \max_{\eta : ||\eta|| \leq \epsilon} \ell(f_\theta(X + \eta), Y) \right] \quad (8)$$

Indeed, this surrogate-based approach is pervasive in practice. Madry et al.'s seminal paper on the subject of adversarial training employs this formulation (Madry et al., 2018), which has subsequently been used as the starting point for numerous AT schemes (Huang et al., 2015; Kurakin et al., 2017).

**Pitfalls of surrogate-based optimization.** Despite the intuitive appeal of this paradigm, surrogate-based adversarial attacks are known to overestimate robustness (Mosbach et al., 2018; Croce et al., 2020b; Croce & Hein, 2020), and standard adversarial training algorithms are known to fail against strong attacks. Furthermore, this formulation suffers from pitfalls such as robust overfitting (Rice et al., 2020) and trade-offs between robustness and accuracy (Zhang et al., 2019). To combat these shortcomings, empirical adversarial attacks and defenses have increasingly relied on heuristics such as multiple restarts, variable learning rate schedules (Croce & Hein, 2020), and carefully crafted initializations, resulting in a widening gap between the theory and practice of adversarial learning. In the next section, we argue that these pitfalls can be attributed to the fundamental limitations of (8).

## 3 NON-ZERO-SUM FORMULATION OF ADVERSARIAL TRAINING

From an optimization perspective, the surrogate-based approaches to adversarial evaluation and training outlined in § 2.3 engenders two fundamental limitations.

**Limitation I: Weak attackers.** In the adversarial evaluation problem of (7), the adversary maximizes an *upper bound* on the classification error. This means that any solution $\eta^\star$ to (7) is not guaranteed to increase the classification error in (5), resulting in adversaries which are misaligned with the goal of finding adversarial examples. Indeed, when the surrogate is an upper bound on the classification error, the only conclusion about the perturbation $\eta^\star$ obtained from (7) and its *true* objective (5) is:

$$\left\{ \arg\max_{i \in [K]} f_\theta(X + \eta^\star)_i \neq Y \right\} \leq \max_{\eta: ||\eta|| \leq \epsilon} \ell(f_\theta(X + \eta), Y) \tag{9}$$

Notably, the RHS of (9) can be arbitrarily large while the left hand side can simultaneously be equal to zero, i.e., the problem in (7) can fail to produce an adversarial example, even at optimality. Thus, while it is known empirically that attacks based on (7) tend to overestimate robustness (Croce & Hein, 2020), this argument shows that this shortcoming is evident *a priori*.

**Limitation II: Ineffective defenders.** Because attacks which seek to maximize upper bounds on the classification error are not proper surrogates for the classification error (c.f., Limitation I), training a model $f_\theta$ on such perturbations does not guarantee any improvement in robustness. Therefore, AT algorithms which seek to solve (8) are ineffective in that they do not optimize the worst-case classification error. For this reason, it should not be surprising that robust overfitting (Rice et al., 2020) occurs for models trained to solve eq. (8).

Both of Limitation I and Limitation II arise directly by virtue of rewriting (7) and (8) with the surrogate loss $\ell$. To illustrate this more concretely, consider the following example.

**Example 1.** *Let $\epsilon > 0$ be given, let $K$ denote the number of classes in a classification problem, and let $\ell$ denote the cross-entropy loss. Consider two possible logit vectors of class probabilities:*

$$z_A = (1/K + \epsilon, 1/K - \epsilon, 1/K, \ldots, 1/K), \qquad z_B = (0.5 - \epsilon, 0.5 + \epsilon, 0, \ldots, 0) \tag{10}$$

*Assume without loss of generality that the correct class is the first class. Then $z_A$ does not lead to an adversarial example, whereas $z_B$ does. However, observe that $\ell(z_A, 1) = -\log(1/K + \epsilon)$, which tends to $\infty$ as $K \to \infty$ and $\epsilon \to 0$. In contrast, $\ell(z_B, 1) = -\log(0.5 - \epsilon)$ which remains bounded as $\epsilon \to 0$. Hence, an adversary maximizing the cross-entropy will always choose $z_A$ over $z_B$ and will therefore fail to identify the adversarial example.*

Therefore, to summarize, there is a distinct tension between the efficient, yet misaligned paradigm of surrogate-based adversarial training with the principled, yet intractable paradigm of minimax optimization on the classification error. In the remainder of this section, we resolve this tension by decoupling the optimization problems of the attacker and the defender.

### 3.1 DECOUPLING ADVERSARIAL ATTACKS AND DEFENSES

Our starting point is the two-player zero-sum formulation in (6). Observe that this minimax optimization problem can be equivalently cast as a *bilevel* optimization problem[1]:

$$\min_{\theta \in \Theta} \quad \mathbb{E}\left\{ \arg\max_{i \in [K]} f_\theta(X + \eta^\star)_i \neq Y \right\} \tag{11}$$

$$\text{subject to} \quad \eta^\star \in \arg\max_{\eta: ||\eta|| \leq \epsilon} \left\{ \arg\max_{i \in [K]} f_\theta(X + \eta)_i \neq Y \right\} \tag{12}$$

While this problem still constitutes a zero-sum game, the role of the attacker (the constraint in (12)) and the role of the defender (the objective in (11)) are now decoupled. From this perspective, the tension engendered by introducing surrogate losses is laid bare: the attacker ought to maximize a *lower bound* on the classification error (c.f., Limitation I), whereas the defender ought to minimize an *upper bound* on the classification error (c.f., Limitation II). This implies that to preserve guarantees

---

[1]To be precise, the optimal value $\eta^\star$ in (17) is a function of $(X, Y)$, i.e., $\eta^\star = \eta^\star(X, Y)$, and the constraint must hold for almost every $(X, Y) \sim \mathcal{D}$. We omit these details for ease of exposition.

on optimality, the attacker and defender must optimize separate objectives. In what follows, we discuss these objectives for the attacker and defender in detail.

**The attacker's objective.** We first address the role of the attacker. To do so, we define the *negative margin* $M_\theta(X, Y)$ of the classifier $f_\theta$ as follows:

$$M_\theta : \mathcal{X} \times \mathcal{Y} \to \mathbb{R}^k, \qquad M_\theta(X, Y)_j \triangleq f_\theta(X)_j - f_\theta(X)_Y \tag{13}$$

We call $M_\theta(X, Y)$ the negative margin because a positive value of (13) corresponds to a misclassification. As we show in the following proposition, the negative margin function (which is differentiable) provides an alternative characterization of the classification error.

**Proposition 1.** *Given a fixed data pair $(X, Y)$, let $\eta^\star$ denote any maximizer of $M_\theta(X + \eta, Y)_j$ over the classes $j \in [K] - \{Y\}$ and perturbations $\eta \in \mathbb{R}^d$ satisfying $||\eta|| \le \epsilon$, i.e.,*

$$(j^\star, \eta^\star) \in \underset{j \in [K] - \{Y\},\, \eta : ||\eta|| \le \epsilon}{\arg\max} M_\theta(X + \eta, Y)_j. \tag{14}$$

*Then if $M_\theta(X + \eta^\star, Y)_{j^\star} > 0$, $\eta^\star$ induces a misclassification and satisfies the constraint in (12), meaning that $X + \eta^\star$ is an adversarial example. Otherwise, if $M_\theta(X + \eta^\star, Y)_{j^\star} \le 0$, then any $\eta : ||\eta|| < \epsilon$ satisfies (12), and no adversarial example exists for the pair $(X, Y)$. In summary, if $\eta^\star$ is as in eq. (14), then $\eta^\star$ solves the lower level problem in eq. (12).*

We present a proof in appendix A[2]. Proposition 1 implies that the non-differentiable constraint in (12) can be equivalently recast as an ensemble of $K$ differentiable optimization problems that can be solved independently. This can collectively be expressed as

$$\eta^\star \in \underset{\eta : ||\eta|| < \epsilon}{\arg\max}\, \underset{j \in [K] - \{Y\}}{\max} M_\theta(X + \eta, Y)_j. \tag{15}$$

Note that this does not constitute a relaxation; (12) and (15) are equivalent optimization problems. This means that the attacker can maximize the classification error directly using first-order optimization methods without resorting to a relaxation. Furthermore, in Appendix D, we give an example of a scenario wherein solving (15) retrieves the optimal adversarial perturbation whereas maximizing the standard adversarial surrogate fails to do so.

**The defender's objective.** Next, we consider the role of the defender. To handle the discontinuous upper-level problem in (11), note that this problem is equivalent to a perturbed version of the supervised learning problem in (1). As discussed in § 2.1, the strongest results for problems of this kind have historically been achieved by means of a surrogate-based relaxation. Subsequently, replacing the 0-1 loss with a differentiable upper bound like the cross-entropy is a principled, guarantee-preserving approach for the defender.

## 3.2 Putting the pieces together: Non-zero-sum adversarial training

By combining the disparate problems discussed in the preceeding section, we arrive at a novel *non-zero-sum* (almost-everywhere) differentiable formulation of adversarial training:

$$\min_{\theta \in \Theta} \quad \mathbb{E}\, \ell(f_\theta(X + \eta^\star), Y) \tag{16}$$

$$\text{subject to} \quad \eta^\star \in \underset{\eta : ||\eta|| \le \epsilon}{\arg\max}\, \underset{j \in [K] - \{Y\}}{\max} M_\theta(X + \eta, y)_j \tag{17}$$

Notice that the second level of this bilevel problem remains non-smooth due to the maximization over the classes $j \in [K] - \{Y\}$. To impart smoothness on the problem without relaxing the constraint, observe that we can equivalently solve $K - 1$ distinct smooth problems in the second level for each sample $(X, Y)$, resulting in the following equivalent optimization problem:

$$\min_{\theta \in \Theta} \quad \mathbb{E}\, \ell(f_\theta(X + \eta^\star_{j^\star}), Y) \tag{18}$$

$$\text{subject to} \quad \eta^\star_j \in \underset{\eta : ||\eta|| \le \epsilon}{\arg\max}\, M_\theta(X + \eta, y)_j \qquad \forall j \in [K] - \{Y\} \tag{19}$$

$$j^\star \in \underset{j \in [K] - \{Y\}}{\arg\max}\, M_\theta(x + \eta^\star_j, y)_j \tag{20}$$

---

[2]This result is similar in spirit to (Gowal et al., 2019, Theorem 3.1). However, (Gowal et al., 2019, Theorem 3.1) only holds for linear functions, whereas Proposition 1 holds for an arbitrary function $f_\theta$.

---

**Algorithm 1:** Best Targeted Attack (BETA)

---

**Input:** Data-label pair $(x, y)$, perturbation size $\epsilon$, model $f_\theta$, number of classes $K$, iterations $T$
**Output:** Adversarial perturbation $\eta^\star$
1 **function** BETA$(x, y, \epsilon, f_\theta, T)$
2     **for** $j \in 1, \ldots, K$ **do**
3         $\eta_j \leftarrow \text{Unif}[\max(X - \epsilon, 0), \min(X + \epsilon, 1)]$          `// Images are in [0,1]`$^d$
4     **for** $t = 1, \ldots, T$ **do**
5         **for** $j \in 1, \ldots, K$ **do**
6             $\eta_j \leftarrow \text{OPTIM}(\eta_j, \nabla_{\eta_i} M_\theta(x + \eta_j, y)_j)$          `// Optimization step`
7             $\eta_j \leftarrow \Pi_{B_\epsilon(X) \cap [0,1]^d}(\eta_j)$          `// Project onto perturbation set`
8     $j^\star \leftarrow \arg\max_{j \in [K] - \{y\}} M_\theta(x + \eta_j, y)$
9     **return** $\eta_{j^\star}$

---

**Algorithm 2:** BETA Adversarial Training (BETA-AT)

---

**Input:** Dataset $(X, Y) = (x_i, y_i)_{i=1}^n$, perturbation size $\epsilon$, model $f_\theta$, number of classes $K$,
        iterations $T$, attack iterations $T'$
**Output:** Robust model $f_{\theta^\star}$
1 **function** BETA-AT$(X, Y, \epsilon, f_\theta, T, T')$
2     **for** $t \in 1, \ldots, T$ **do**
3         Sample $i \sim \text{Unif}[n]$
4         $\eta^\star \leftarrow \text{BETA}(x_i, y_i, \epsilon, f_\theta, T')$
5         $L(\theta) \leftarrow \ell(f_\theta(x_i + \eta^\star), y_i)$
6         $\theta \leftarrow \text{OPTIM}(\theta, \nabla L(\theta))$          `// Optimization step`
7     **return** $f_\theta$

---

Hence, in (20), we first obtain one perturbation $\eta_j^\star$ per class which maximizes the negative margin $M_\theta(X + \eta_j^\star, Y)$ for that particular class. Next, in (19), we select the class index $j^\star$ corresponding to the perturbation $\eta_j^\star$ that maximized the negative margin. And finally, in the upper level, the surrogate minimization over $\theta \in \Theta$ is on the perturbed data pair $(X + \eta_{j^\star}^\star, Y)$. The result is a non-zero-sum formulation for AT that is amenable to gradient-based optimization, and preserves the optimality guarantees engendered by surrogate loss minimization without weakening the adversary.

## 4   Algorithms

Given the non-zero-sum formulation of AT, the next question is how one should solve this bilevel problem in practice. Our starting point is the empirical version of this bilevel problem, wherein we assume access to a finite dataset $\{(x_i, y_i)\}_{i=1}^n$ of $n$ instance-label pairs sampled i.i.d. from $\mathcal{D}$.

$$\min_{\theta \in \Theta} \quad \frac{1}{n} \sum_{i=1}^n \ell(f_\theta(x_i + \eta_{ij^\star}^\star), y_i) \tag{21}$$

$$\text{subject to} \quad \eta_{ij}^\star \in \arg\max_{\eta : \|\eta\| \leq \epsilon} M_\theta(x_i + \eta, y_i)_j \qquad \forall i, j \in [n] \times [K] - \{Y\} \tag{22}$$

$$j^\star \in \arg\max_{j \in [K] - \{y_i\}} M_\theta(x_i + \eta_{ij}^\star, y_i)_j \qquad \forall i \in [n] \tag{23}$$

To solve this empirical problem, we adopt a stochastic optimization based approach. That is, we first iteratively sample mini-batches from our dataset uniformly at random, and then obtain adversarial perturbations by solving the lower level problems in (22) and (23). Note that given the differentiability of the negative margin, the lower level problems can be solved iteratively with generic optimizers, e.g., Adam (Kingma & Ba, 2014) or RMSprop. This procedure is summarized in Algorithm 1, which we call the **BE**st **T**argeted **A**ttack (BETA), given that it directly maximizes the classification error.

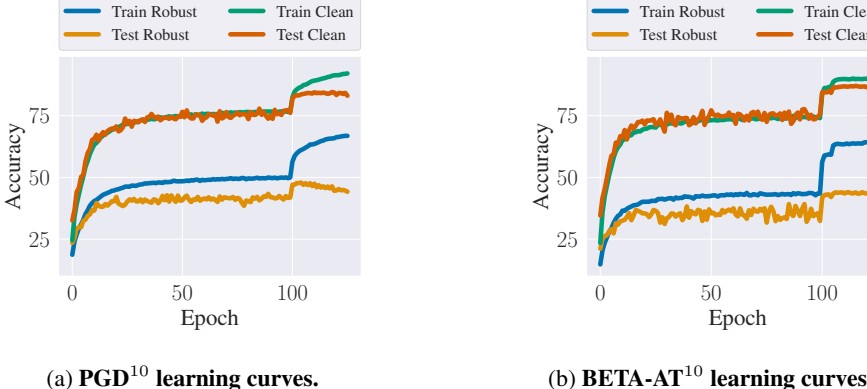

(a) **PGD**[10] **learning curves.**                    (b) **BETA-AT**[10] **learning curves.**

Figure 1: **BETA does not suffer from robust overfitting.** We plot the learning curves against a PGD[20] adversary for PGD[10] and BETA-AT[10]. Observe that although PGD displays robust overfitting after the first learning rate decay step, BETA-AT does not suffer from this pitfall.

After obtaining such perturbations, we calculate the perturbed loss in (21), and then differentiate through this loss with respect to the model parameters. By updating the model parameters $\theta$ in the negative direction of this gradient, our algorithm seeks classifiers that are robust against perturbations found by BETA. We call the full adversarial training procedure based on this attack *BETA Adversarial Training (BETA-AT)*, as it invokes BETA as a subroutine; see Algorithm 2 for details. Also see Figures 2 and 3 in the appendix for an empirical study of the computational complexity of BETA.

## 5 EXPERIMENTS

In this section, we evaluate the performance of BETA and BETA-AT on CIFAR-10 (Krizhevsky et al., 2009). Throughout, we consider a range of AT algorithms, including PGD (Madry et al., 2018), FGSM (Goodfellow et al., 2015), TRADES (Zhang et al., 2019), MART (Wang et al., 2020), as well as a range of adversarial attacks, including APGD and AutoAttack (Croce & Hein, 2020). We consider the standard perturbation budget of $\epsilon = 8/255$, and all training and test-time attacks use a step size of $\alpha = 2/255$. For both TRADES and MART, we set the trade-off parameter $\lambda = 5$, which is consistent with the original implementations (Wang et al., 2020; Zhang et al., 2019).

**The bilevel formulation eliminates robust overfitting.** Robust overfitting occurs when the robust test accuracy peaks immediately after the first learning rate decay, and then falls significantly in subsequent epochs as the model continues to train (Rice et al., 2020). This is illustrated in Figure 1a, in which we plot the learning curves (i.e., the clean and robust accuracies for the training and test sets) for a ResNet-18 (He et al., 2016) trained using 10-step PGD against a 20-step PGD adversary. Notice that after the first learning rate decay at epoch 100, the robust test accuracy spikes, before dropping off in subsequent epochs. On the other hand, BETA-AT does not suffer from robust overfitting, as shown in Figure 1b. We argue that this strength of our method is a direct result of our bilevel formulation, in which we train against a proper surrogate for the adversarial classification error.

**BETA-AT outperforms baselines on the last iterate of training.** We next compare the performance of ResNet-18 models trained using four different AT algorithms: FGSM, PGD, TRADES, MART, and BETA. PGD, TRADES, and MART used a 10-step adversary at training time. At test time, the models were evaluated against five different adversaries: FGSM, 10-step PGD, 40-step PGD, 10-step BETA, and APGD. We report the performance of two different checkpoints for each algorithm: the best performing checkpoint chosen by early stopping on a held-out validation set, and the performance of the last checkpoint from training. Note that while BETA performs comparably to the baseline algorithms with respect to early stopping, it outperforms these algorithms significantly when the test-time adversaries attack the last checkpoint of training. This owes to the fact that BETA does not suffer from robust overfitting, meaning that the last and best checkpoints perform similarly.

**BETA matches the performance of AutoAttack.** AutoAttack is a state-of-the-art attack which is widely used to estimate the robustness of trained models on leaderboards such as RobustBench (Croce

Table 1: **Adversarial performance on CIFAR-10.** We report the test accuracies of various AT algorithms against different adversarial attacks on the CIFAR-10 dataset.

| Training algorithm | Test accuracy | | | | | | | | | | | |
|---|---|---|---|---|---|---|---|---|---|---|---|---|
| | Clean | | FGSM | | $\text{PGD}^{10}$ | | $\text{PGD}^{40}$ | | $\text{BETA}^{10}$ | | APGD | |
| | Best | Last | Best | Last | Best | Last | Best | Last | Best | Last | Best | Last |
| FGSM | 81.96 | 75.43 | 94.26 | 94.22 | 42.64 | 1.49 | 42.66 | 1.62 | 40.30 | 0.04 | 41.56 | 0.00 |
| $\text{PGD}^{10}$ | 83.71 | 83.21 | 51.98 | 47.39 | 46.74 | 39.90 | 45.91 | 39.45 | 43.64 | 40.21 | 44.36 | 42.62 |
| $\text{TRADES}^{10}$ | 81.64 | 81.42 | 52.40 | 51.31 | 47.85 | 42.31 | 47.76 | 42.92 | 44.31 | 40.97 | 43.34 | 41.33 |
| $\text{MART}^{10}$ | 78.80 | 77.20 | 53.84 | 53.73 | 49.08 | 41.12 | 48.41 | 41.55 | 44.81 | 41.22 | 45.00 | 42.90 |
| $\text{BETA-AT}^{5}$ | 87.02 | 86.67 | 51.22 | 51.10 | 44.02 | 43.22 | 43.94 | 42.56 | 42.62 | 42.61 | 41.44 | 41.02 |
| $\text{BETA-AT}^{10}$ | 85.37 | 85.30 | 51.42 | 51.11 | 45.67 | 45.39 | 45.22 | 45.00 | 44.54 | 44.36 | 44.32 | 44.12 |
| $\text{BETA-AT}^{20}$ | 82.11 | 81.72 | 54.01 | 53.99 | 49.96 | 48.67 | 49.20 | 48.70 | 46.91 | 45.90 | 45.27 | 45.25 |

Table 2: **Estimated $\ell_\infty$ robustness (robust test accuracy).** BETA+RMSprop (ours) vs APGD-targeted (APGD-T) vs AutoAttack (AA). CIFAR-10. BETA and APGD-T use 30 iterations + single restart. $\epsilon = 8/255$. AA uses 4 different attacks with 100 iterations and 5 restarts.

| Model | BETA | APGD-T | AA | BETA/AA gap | Architecture |
|---|---|---|---|---|---|
| Wang et al. (2023) | 70.78 | 70.75 | 70.69 | 0.09 | WRN-70-16 |
| Wang et al. (2023) | 67.37 | 67.33 | 67.31 | 0.06 | WRN-28-10 |
| Rebuffi et al. (2021) | 66.75 | 66.71 | 66.58 | 0.17 | WRN-70-16 |
| Gowal et al. (2021) | 66.27 | 66.26 | 66.11 | 0.16 | WRN-70-16 |
| Huang et al. (2022) | 65.88 | 65.88 | 65.79 | 0.09 | WRN-A4 |
| Rebuffi et al. (2021) | 64.73 | 64.71 | 64.64 | 0.09 | WRN-106-16 |
| Rebuffi et al. (2021) | 64.36 | 64.27 | 64.25 | 0.11 | WRN-70-16 |
| Gowal et al. (2021) | 63.58 | 63.45 | 63.44 | 0.14 | WRN-28-10 |
| Pang et al. (2022) | 63.38 | 63.37 | 63.35 | 0.03 | WRN-70-16 |

et al., 2020a; Croce & Hein, 2020). In brief, AutoAttack comprises a collection of four disparate attacks: APGD-CE, APGD-T, FAB, and Square Attack. AutoAttack also involves several heuristics, including multiple restarts and variable stopping conditions. In Table 2, we compare the performance of the top-performing models on RobustBench against AutoAttack, APGD-T, and BETA with RMSprop. Both APGD-T and BETA used thirty steps, whereas we used the default implementation of AutoAttack, which runs for 100 iterations. We also recorded the gap between AutoAttack and BETA. Notice that the 30-step BETA—a heuristic-free algorithm derived from our bilevel formulation of AT—performs almost identically to AutoAttack, despite the fact that AutoAttack runs for significantly more iterations and uses five restarts, which endows AutoAttack with an unfair computational advantage. That is, excepting for a negligible number of samples, BETA matches the performance of AutoPGD-targeted and AutoAttack, despite using an off-the-shelf optimizer.

# 6 RELATED WORK

**Robust overfitting.** Several recent papers (see, e.g., (Rebuffi et al., 2021; Chen et al., 2021; Yu et al., 2022; Dong et al., 2022; Wang et al., 2020; Lee et al., 2020)) have attempted to explain and resolve robust overfitting (Rice et al., 2020). However, none of these works point to a fundamental limitation of AT as the cause of robust overfitting. Rather, much of this past work has focused on proposing heuristics for algorithms specifically designed to reduce robust overfitting, rather than to improve AT. In contrast, we posit that the lack of guarantees of the zero-sum surrogate-based AT paradigm Madry et al. (2018) is at fault, as this paradigm is not designed to maximize robustness with respect to classification error. And indeed, our empirical evaluations in the previous section confirm that our non-zero-sum formulation eliminates robust overfitting.

**Estimating adversarial robustness.** There is empirical evidence that attacks based on surrogates (e.g., PGD) overestimate the robustness of trained classifiers (Croce & Hein, 2020; Croce et al., 2020b). Indeed, this evidence served as motivation for the formulation of more sophisticated attacks like AutoAttack (Croce & Hein, 2020), which tend to provide more accurate estimates of robustness. In contrast, we provide solid, theoretical evidence that commonly used attacks overestimate robustness due to the misalignment between standard surrogate losses and the adversarial classification error. Moreover, we show that optimizing the BETA objective with a standard optimizer (e.g., RMSprop)

achieves the same robustness as AutoAttack without employing ad hoc training procedures such as multiple restarts. convoluted stopping conditions, or adaptive learning rates.

One notable feature of past work is an overservation made in (Gowal et al., 2019), which finds that multitargeted attacks tend to more accurately estimate robustness. However, their theoretical analysis only applies to linear functions, whereas our work extends these ideas to the nonlinear setting of DNNs. Moreover, (Gowal et al., 2019) do not explore *training* using a multitargeted attack, whereas we show that BETA-AT is an effective AT algorithm that mitigates the impact of robust overfitting.

**Bilevel formulations of AT.** Prior to our work, (Zhang et al., 2022) proposed a different *pseudo-bilevel*[3] formulation for AT, wherein the main objective was to justify the FastAT algorithm introduced in (Wong et al., 2020). Specifically, the formulation in (Zhang et al., 2022) is designed to produce solutions that coincide with the iterates of FastAT by linearizing the attacker's objective. In contrast, our bilevel formulation appears naturally following principled relaxations of the intractable classification error AT formulation. In this way, the formulation in (Zhang et al., 2022) applies only in the context of Fast AT, whereas our formulation deals more generally with the task of AT.

In the same spirit as our work, (Mianjy & Arora, 2024) solve a problem equivalent to a bilevel problem wherein the adversary maximizes a "reflected" cross-entropy loss. While this paper focuses on binary classification, the authors show that this approach leads to improved adversarial robustness and admits convergence guarantees. Our approach, while related, is distinct in its reformulation of the adversarial training problem via the negative margin loss. Moreover, our results show that BETA mitigates robustness overfitting and is roughly five times as effective as AutoAttack.

**Theoretical underpinnings of surrogate minimization.** In this paper, we focused on the *empirical* performance of AT in the context of the literature concerning adversarial examples in computer vision. However, the study of the efficacy of surrogate losses in minimizing the target 0-1 loss is a well studied topic among theorists. Specifically, this literature considers two notions of minimizers for the surrogate loss also minimizing the target loss: (1) consistency, which requires uniform convergence, and (2) calibration, which requires the weaker notion of pointwise convergence (although (Bartlett et al., 2006) shows that these notions are equivalent for standard, i.e., non-adversarial, classification).

In the particular case of classification in the presence of adversaries, (Bao et al., 2020) and (Meunier et al., 2022) claimed that for the class of linear models, no convex surrogate loss is calibrated with respect to the 0-1 zero-sum formulation of AT, although certain classes of nonconvex losses can maintain calibration for such settings. However, in (Awasthi et al., 2021), the authors challenge this claim, and generalize the calibration results considered by (Bao et al., 2020) beyond linear models. One interesting direction future work would be to provide a theoretical analysis of BETA with respect to the margin-based consistency results proved very recently in (Frank & Niles-Weed, 2023). We also note that in parallel, efforts have been made to design algorithms that are approximately calibrated, leading to—among other things—the TRADES algorithm (Zhang et al., 2019), which we compare to in Section 5. Our work is in the same vein, although BETA does not require approximating a divergence term, which leads to non-calibration of the TRADES objective.

## 7 CONCLUSION

In this paper, we argued that the surrogate-based relaxation commonly employed to improve the tractability of adversarial training voids guarantees on the ultimate robustness of trained classifiers, resulting in weak adversaries and ineffective algorithms. This shortcoming motivated the formulation of a novel, yet natural bilevel approach to adversarial training and evaluation in which the adversary and defender optimize separate objectives, which constitutes a non-zero-sum game. Based on this formulation, we developed a new adversarial attack algorithm (BETA) and a concomitant AT algorithm, which we call BETA-AT. In our experiments, we showed that BETA-AT eliminates robust overfitting and we showed that even when early stopping based model selection is used, BETA-AT performs comparably to AT. Finally, we showed that BETA provides almost identical estimates of robustness to AutoAttack.

---

[3]In a strict sense, the formulation of (Zhang et al., 2022) is not a bilevel problem. In general, the most concise way to write a bilevel optimization problem is $\min_\theta f(\theta, \delta^\star(\theta))$ subject to $\delta^\star(\theta) \in \arg\max g(\theta, \delta)$. In such problems the value $\delta^\star(\theta)$ only depends on $\theta$, as the objective function $g(\theta, \cdot)$ is then uniquely determined. This is not the case in (Zhang et al., 2022, eq. (7)), where an additional variable $z$ appears, corresponding to the random initialization of Fast-AT. Hence, in (Zhang et al., 2022) the function $g(\theta, \cdot)$ is not uniquely defined by $\theta$, but is a random function realized at each iteration of the algorithm.

ACKNOWLEDGEMENTS

FL is funded (in part) through a PhD fellowship of the Swiss Data Science Center, a joint venture between EPFL and ETH Zurich. VC is supported by the Hasler Foundation Program: Hasler Responsible AI (project number 21043), the Army Research Office under grant number W911NF-24-1-0048, and the Swiss National Science Foundation (SNSF) under grant number 200021-205011. AR, HH, and GP are supported by the NSF Institute for CORE Emerging Methods in Data Science (EnCORE). AR is also supposed by an ASSET Amazon AWS Trustworthy AI Fellowship.

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

# Appendices

## Table of Contents

## A  PROOF OF PROPOSITION 1

Suppose that there exists $\hat{\eta}$ satisfying $||\hat{\eta}|| \leq \epsilon$ such that for some $j \in [K]$, $j \neq Y$ we have $M_\theta(X + \hat{\eta}, Y)_j > 0$. That is, assume that

$$\max_{j \in [K] - \{Y\}, \, \eta: \|\eta\| \leq \epsilon} M_\theta(X + \eta, Y)_j > 0 \tag{24}$$

and for some $\hat{\eta}$ and some $j$ we have $f_\theta(X + \hat{\eta})_j > f_\theta(X + \hat{\eta})_Y$, which implies that $\arg\max_{j \in [K]} f_\theta(X + \hat{\eta})_j \neq Y$. Hence, $\hat{\eta}$ induces a misclassification error, i.e.,

$$\hat{\eta} \in \arg\max_{\eta: \|\eta\|_2 \leq \epsilon} \left\{ \arg\max_{j \in [K]} f_\theta(X + \eta)_j \neq Y \right\}. \tag{25}$$

In particular, if

$$(j^\star, \eta^\star) \in \arg\max_{j \in [K] - \{Y\}, \, \eta: \|\eta\| \leq \epsilon} M_\theta(X + \eta, Y)_j \tag{26}$$

then it holds that

$$\eta^\star \in \arg\max_{\eta: \|\eta\|_2 \leq \epsilon} \left\{ \arg\max_{j \in [K]} f_\theta(X + \eta)_j \neq Y \right\}. \tag{27}$$

Otherwise, if it holds that

$$\max_{j \in [K] - \{Y\}, \, \eta: \|\eta\| \leq \epsilon} M_\theta(X + \eta, Y)_j < 0, \tag{28}$$

then for all $\eta : \|\eta\| < \epsilon$ and all $j \neq Y$, we have $f_\theta(X + \eta)_j < f_\theta(X + \eta)_Y$, so that $\arg\max_{j \in [K]} f_\theta(x + \eta)_j = Y$, i.e., there is no adversarial example in the ball. In this case, for any $\eta$, if it holds that

$$(j^\star, \eta^\star) \in \arg\max_{j \in [K] - \{Y\}, \, \eta: \|\eta\| \leq \epsilon} M_\theta(X + \eta, Y)_j \tag{29}$$

then

$$0 = \left\{ \arg\max_{j \in [K]} f_\theta(X + \eta^\star)_j \neq Y \right\} = \max_{\eta: \|\eta\|_2 \leq \epsilon} \left\{ \arg\max_{j \in [K]} f_\theta(X + \eta)_j \neq Y \right\} \tag{30}$$

In conclusion, the solution

$$(j^\star, \eta^\star) \in \arg\max_{j \in [K] - \{Y\}, \, \eta: \|\eta\| \leq \epsilon} M_\theta(X + \eta, Y)_j \tag{31}$$

always yields a maximizer of the misclassification error.

---

**Algorithm 3:** Smooth BETA Adversarial Training (SBETA-AT)

**Input:** Dataset $(X, Y) = (x_i, y_i)_{i=1}^n$, perturbation size $\epsilon$, model $f_\theta$, number of classes $K$, iterations $T$, attack iterations $T'$, temperature $\mu > 0$

**Output:** Robust model $f_{\theta^*}$

1 **function** SBETA-AT($X, Y, \epsilon, \theta, T, \gamma, \mu$)
2    **for** $t \in 1, \ldots, T$ **do**
3      Sample $i \sim \text{Unif}[n]$
4      Initialize $\eta_j \sim \text{Unif}[\max(0, x_i - \epsilon), \min(x_i + \epsilon, 1)], \forall j \in [K]$
5      **for** $j \in 1, \ldots, K$ **do**
6        **for** $t \in 1, \ldots, T'$ **do**
7          $\eta_j \leftarrow \text{OPTIM}(\eta_j, \nabla_\eta M_\theta(x_i + \eta_j, y_i)_j)$ ▷ (attack optimizer step, e.g., RMSprop)
8          $\eta_j \leftarrow \Pi_{B_\epsilon(x_i) \cap [0,1]^d}(\eta_j)$            ▷ (Projection onto valid perturbation set)
9      Compute $L(\theta) = \sum_{j=1, j \neq y_i}^K \frac{e^{\mu M_\theta(x_i + \eta_j, y_i)_j}}{\sum_{j=1, j \neq y_i}^K e^{\mu M_\theta(x_i + \eta_j, y_i)_j}} \ell(f_\theta(x_i + \eta_j), y_i)$
10      $\theta \leftarrow \text{OPTIM}(\theta, \nabla L(\theta))$                      ▷ (model optimizer step)
11    **return** $f_\theta$

---

# B   SMOOTH REFORMULATION OF THE LOWER LEVEL

First, note that the problem in eqs. (21) to (23) is equivalent to

$$\min_{\theta \in \Theta} \frac{1}{n} \sum_{i=1}^n \sum_{j=1}^K \lambda_{ij}^\star \ell(f_\theta(x_i + \eta_{ij}^\star), y_i)$$

$$\text{subject to } \lambda_{ij}^\star, \eta_{ij}^\star \in \underset{\substack{\|\eta_{ij}\| \leq \epsilon \\ \lambda_{ij} \geq 0, \|\lambda_i\|_1 = 1, \lambda_{iy} = 0}}{\arg\max} \sum_{j=1}^K \lambda_{ij} M_\theta(x_i + \eta_{ij}, y_i)_j \qquad \forall i \in [n] \tag{32}$$

This is because the maximum over $\lambda_i$ in eq. (32) is always attained at the coordinate vector $\mathbf{e}_j$ such that $M_\theta(x_i + \eta_{ij}^\star, y_i)$ is maximum.

An alternative is to smooth the lower level optimization problem by adding an entropy regularization:

$$\max_{\eta: \|\eta\| \leq \epsilon} \max_{j \in [K] - \{y\}} M_\theta(x + \eta_j, y)_j = \max_{\eta: \|\eta\| \leq \epsilon} \max_{\lambda \geq 0, \|\lambda\|_1 = 1, \lambda_y = 0} \langle \lambda, M_\theta(x + \eta_j, y)_{j=1}^K \rangle$$

$$\geq \max_{\eta: \|\eta\| \leq \epsilon} \max_{\lambda \geq 0, \|\lambda\|_1 = 1, \lambda_y = 0} \langle \lambda, M_\theta(x + \eta_j, y)_{j=1}^K \rangle - \frac{1}{\mu} \sum_{j=1}^K \lambda_j \log(\lambda_j)$$

$$= \max_{\eta: \|\eta\| \leq \epsilon} \frac{1}{\mu} \log \left( \sum_{\substack{j=1 \\ j \neq y}}^K e^{\mu M_\theta(X + \eta, y)_j} \right) \tag{33}$$

where $\mu > 0$ is some *temperature* constant. The inequality here is due to the fact that the entropy of a discrete probability $\lambda$ is positive. The innermost maximization problem in (33) has the closed-form solution:

$$\lambda_j^\star = \frac{e^{\mu M_\theta(x + \eta_j, y)_j}}{\sum_{\substack{j=1 \\ j \neq y}}^K e^{\mu M_\theta(x + \eta_j, y)_j}} : j \neq y, \qquad \lambda_y^\star = 0 \tag{34}$$

Hence, after relaxing the second level maximization problem following eq. (33), and plugging in the optimal values for $\lambda$ we arrive at:

$$\min_{\theta \in \Theta} \frac{1}{n} \sum_{i=1}^{n} \sum_{\substack{j=1 \\ j \neq y_i}}^{K} \frac{e^{\mu M_\theta(x_i + \eta_{ij}, y_i)_j}}{\sum_{\substack{j=1 \\ j \neq y_i}}^{K} e^{\mu M_\theta(x_i + \eta_{ij}, y_i)_j}} \ell(f_\theta(x_i + \eta_{ij}^\star), y_i) \tag{35}$$

$$\text{subject to } \eta_{ij}^\star \in \underset{\|\eta_{ij}\| \leq \epsilon}{\arg\max} \, M_\theta(x_i + \eta_{ij}, y_i)_j \qquad \forall i \in [n], j \in [K]$$

$$\min_{\theta \in \Theta} \quad \frac{1}{n} \sum_{i=1}^{n} \sum_{\substack{j=1 \\ j \neq y_i}}^{K} \frac{e^{\mu M_\theta(x_i + \eta_{ij}^\star, y_i)_j}}{\sum_{\substack{j=1 \\ j \neq y_i}}^{K} e^{\mu M_\theta(x_i + \eta_{ij}^\star, y_i)_j}} \ell(f_\theta(x_i + \eta_{ij}^\star), y_i) \tag{36}$$

$$\text{subject to} \quad \eta_{ij}^\star \in \underset{\eta:\|\eta\| \leq \epsilon}{\arg\max} \, M_\theta(x_i + \eta, y_i)_j \qquad \qquad \forall i \in [n] \tag{37}$$

In this formulation, both upper- and lower-level problems are smooth (barring the possible use of nonsmooth components like ReLU). Most importantly (I) the smoothing is obtained through a lower bound of the original objective in eqs. (22) and (23), retaining guarantees that the adversary will increase the misclassification error and (II) all the adversarial perturbations obtained for each class now appear in the upper level (36), weighted by their corresponding negative margin. In this way, we make efficient use of all perturbations generated: if two perturbations from different classes achieve the same negative margin, they will affect the upper-level objective in fair proportion. This formulation gives rise to algorithm 3.

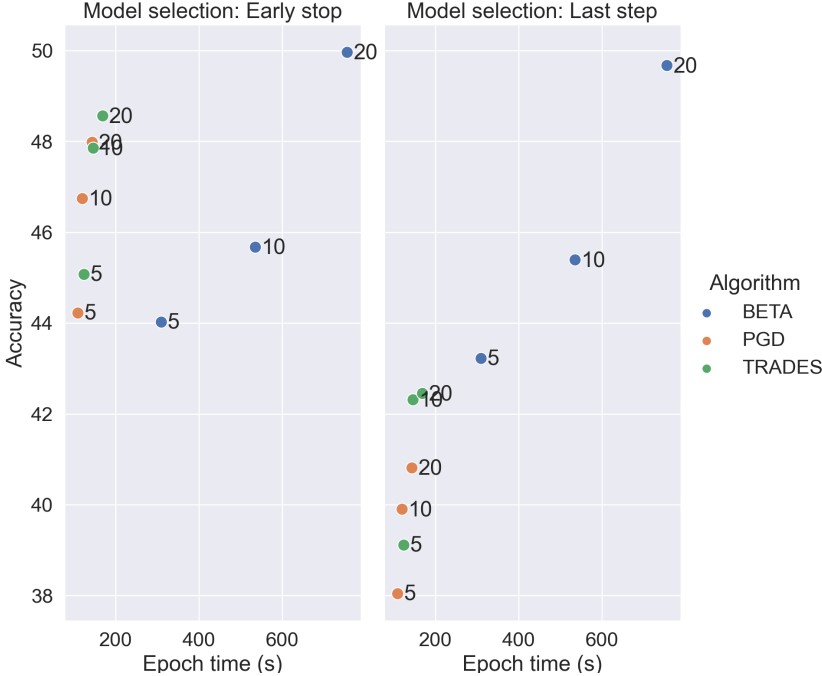

Figure 2: **Adversarial training performance-speed trade-off.** Each point is annotated with the number of steps with which the corresponding algorithm was run. Observe that robust overfitting is eliminated by BETA, but that this comes at the cost of increased computational overhead. This reveals an expected performance-speed trade-off for our algorithm.

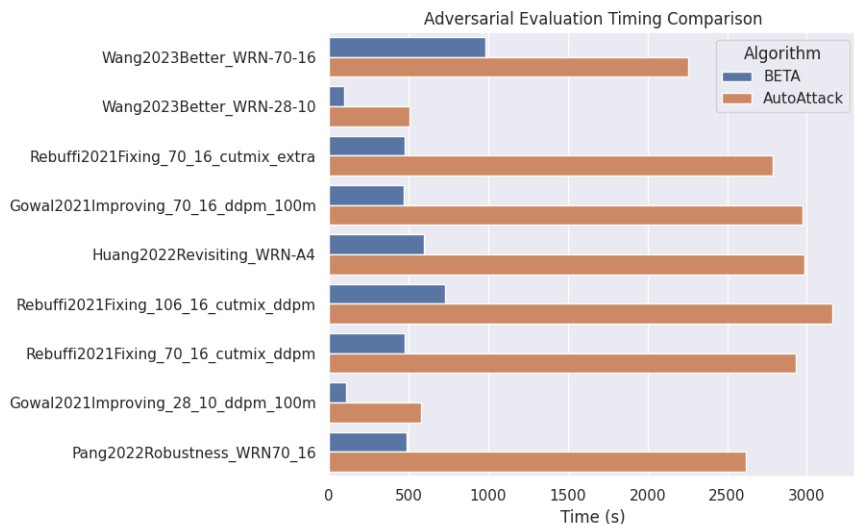

Figure 3: **Adversarial evaluation timing comparison.** The running time for evaluating the top models on RobustBench using AutoAttack and BETA with the same settings as Table 2 are reported. On average, BETA is 5.11 times faster than AutoAttack.

## C  RUNNING TIME ANALYSIS

### C.1  SPEED-PERFORMANCE TRADE-OFF

In Figure 2, we analyze the trade-off between the running time and performance of BETA. Specifically, on the horizontal axis, we plot the running time (in seconds) of an epoch of BETA, and on the vertical axis we plot the performance measured via the robust accuracy with respect to a 20-step PGD adversary. We compare BETA to PGD and TRADES, and we show the speed-performance trade-off when each of these algorithms are run for 5, 10, and 20 iterations; the iteration count is labeled next to each data point. The leftmost panel shows early stopping model selection, and the rightmost panel shows last iterate model selection. Notice that while BETA is significantly more resource intensive than PGD and TRADES, BETA tends to outperform the baselines, particularly if one looks at the gap between early stopping and last iterate model selection.

### C.2  EVALUATION RUNNING TIME ANALYSIS

We next analyze the running time of BETA when used as to adversarially evaluate state-of-the-art robust models. In particular, we return to the setting of Table 2, wherein we compared the performance of AutoAttack to BETA. In Figure 3, we show the wall-clock time of performing adversarial evaluation using both of these algorithms. Notice that AutoAttack takes significantly longer to evaluate each of these models, and as we showed in Table 2, this additional time does not yield a better estimate of the robustness of these models. Indeed, by averaging over the scores in Figure 1b, we find that BETA is $5.11\times$ faster than AutoAttack on average.

# D  UTILITY OF MAXIMIZING THE SURROGATE LOSS

In this appendix, we show that there exists cases in which our margin-based inner maximization retrieves the optimal adversarial perturbation while the standard inner max with the surrogate loss fails to do so. In this example, we consider a classification problem in which the classifier $f : \mathbb{R}^2 \rightarrow \mathbb{R}^3$ is linear across three classes $\{1, 2, 3\}$. Specifically we define $f$ in the following way:

$$f(x_1, x_2) = \begin{bmatrix} 0 & -1 \\ -1 & 0 \\ 1 & 0 \end{bmatrix} \begin{bmatrix} x_1 \\ x_2 \end{bmatrix} \tag{38}$$

Furthermore, let $\epsilon = 0.8$, let $(x_1, x_2) = (0, -1)$, and assume without loss of generality that the correct class is $y = 1$. The solution for the maximization of cross-entropy loss is given by:

$$\max_{\|\eta\| \leq 0.8} \ell(f(x + \eta), 1) = \max_{\|\eta\| \leq 0.8} -\log\left(\frac{e^{1-\eta_2}}{e^{1-\eta_2} + e^{-\eta_1} + e^{\eta_1}}\right) \tag{39}$$

where $\ell$ denotes the cross entropy loss. Now observe that by the monotonicity of the logarithm function, this problem on the right-hand-side is equivalent to the following problem:

$$\min_{\|\eta\| \leq 0.8} \frac{e^{1-\eta_2}}{e^{1-\eta_2} + e^{-\eta_1} + e^{\eta_1}} = 1 + \max_{\|\eta\| \leq 0.8} \frac{e^{-\eta_1} + e^{\eta_1}}{e^{1-\eta_2}} \tag{40}$$

$$= \max_{|\eta_1| \leq 0.8} \max_{|\eta_2| \leq \sqrt{0.8^2 - \eta_1^2}} \frac{e^{-\eta_1} + e^{+\eta_1}}{e^{1-\eta_2}} \tag{41}$$

where in the final step we split the problem so that we optimize separately over $\eta_1$ and $\eta_2$. Observe that the inner problem, for which the numerator is constant, satisfies the following:

$$\max_{|\eta_2| \leq \sqrt{0.8^2 - \eta_1^2}} \frac{e^{-\eta_1} + e^{+\eta_1}}{e^{1-\eta_2}} = \min_{|\eta_2| \leq \sqrt{0.8^2 - \eta_1^2}} e^{1-\eta_2} = \min_{|\eta_2| \leq \sqrt{0.8^2 - \eta_1^2}} 1 - \eta_2 \tag{42}$$

As the objective is linear in the rightmost optimization problem, it's clear that $\eta_2^\star = \sqrt{0.8^2 - \eta_1^2}$. Now returning to (41), we substitute $\eta_2^\star$ and are therefore left to solve the following problem:

$$\max_{|\eta_1| \leq 0.8} \frac{e^{-\eta_1} + e^{\eta_1}}{e^{1-\sqrt{0.8^2 - \eta_1^2}}} = \max_{|\eta_1| \leq 0.8} (e^{-\eta_1} + e^{\eta_1}) e^{\sqrt{0.8^2 - \eta_1^2}} \tag{43}$$

$$= \max_{0 \leq \eta_1 \leq 0.8} \frac{e^{-\eta_1} + e^{\eta_1}}{e^{1-\sqrt{0.8^2 - \eta_1^2}}} \tag{44}$$

where in the final step we used the fact that the objective is symmetric in $\eta_1$. By visual inspection, this function achieves its maximum at $\eta_1^\star = 0$ (see Figure 4). Hence, the optimal perturbation obtained via cross-entropy maximization is $\eta^\star = (0, 0.8)$. Therefore,

$$(x_1, x_2) + (\eta_1^\star, \eta_2^\star) = (0, -1) + (0, 0.8) = (0, -0.2)$$

Then, by applying the classifier $f$, we find that

$$f(0, -0.2) = \begin{bmatrix} 0 & -1 \\ -1 & 0 \\ 1 & 0 \end{bmatrix} \begin{bmatrix} 0 \\ 0.2 \end{bmatrix} = \begin{bmatrix} 0.2 \\ 0 \\ 0 \end{bmatrix} \tag{45}$$

This shows that the class assigned to this optimally perturbed example is still the correct class $y = 1$, i.e., the attacker fails to find an adversarial example. In contrast, the main idea in the derivation of the BETA algorithm is to optimize the margins separately for both possible incorrect classes $y = 2$ and $y = 3$. In particular, for the class $y = 2$, BETA solves the following problem:

$$\max_{\|\eta\| \leq 0.8} ([-1, 0] - [0 - 1]) \cdot (x + \eta) \tag{46}$$

The point $\eta = [-0.8, 0.8]/\sqrt{2}$ is optimal for this linear problem. On the other hand, for the class $y = 3$, BETA solves the following problem:

$$\max_{\|\eta\| \leq 0.8} ([1, 0] - [0 - 1]) \cdot (x + \eta) \tag{47}$$

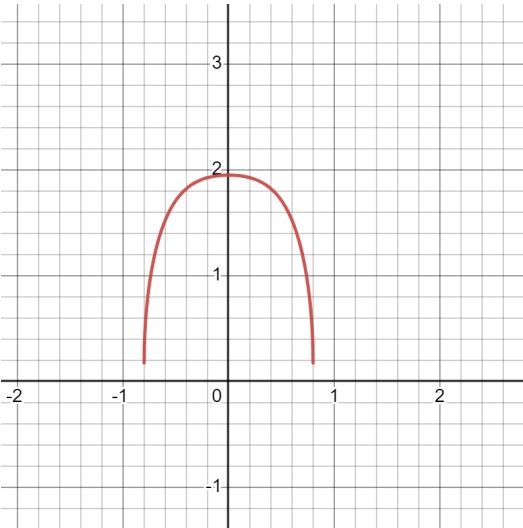

Figure 4: Plot of function to be maximized in eq. (44). We subtract $y = 2.5$ for ease of viewing

The point $\eta = [0.8, 0.8]/\sqrt{2}$ is optimal for this problem. Observe that both achieve the same value of the margin, so BETA can choose either optimal point; without loss of generality, assume that BETA chooses the second point $\eta^\star = [0.8, 0.8]/\sqrt{2}$ as the optimal solution. The corresponding classifier takes the following form:

$$f(0.8/\sqrt{2}, 0.8/\sqrt{2} - 1) = \begin{bmatrix} 0 & -1 \\ -1 & 0 \\ 1 & 0 \end{bmatrix} \begin{bmatrix} 0.8/\sqrt{2} \\ 0.8/\sqrt{2} - 1 \end{bmatrix} \tag{48}$$

$$= \begin{bmatrix} 1 - 0.8/\sqrt{2} \\ -0.8/\sqrt{2} \\ 0.8/\sqrt{2} \end{bmatrix} \tag{49}$$

$$\approx \begin{bmatrix} 0.43 \\ -0.57 \\ 0.57 \end{bmatrix} \tag{50}$$

Hence, the classifier returns the incorrect class, i.e., the attack is successful. This shows that whereas the cross-entropy maximization problem fails to find an adversarial example, BETA succeeds in finding an adversarial example.

