# OpenReview forum: "Adversarial Training Should Be Cast as a Non-Zero-Sum Game"
_ICLR.cc/2024/Conference — ICLR 2024 poster_

### Official Review · Reviewer_NF3x · 2023-11-01

**Soundness:** 3 good
**Presentation:** 3 good
**Contribution:** 2 fair
**Rating:** 6
**Confidence:** 4

**Summary:**

The paper addresses the problem of training a classifier to be robust to adversarial attacks wherein an adversary can arbitrarily perturb a test data point $X$ a neighboring data point $X+\eta$ within a small ball of radius $\epsilon$. A popular way to train such a classifier is Adversarial Training (AT) with a min-max objective where the attacker solves the inner maximization of the loss function and the defender solves the outer minimization of the worst case loss. The paper identifies that solving the inner maximization by replacing the 0-1 misclassification with a surrogate loss can lead to suboptimal adversarial perturbations, and subsequently, the defenders trained on such weak attacks result in suboptimal robust accuracy. Building upon this observation, the paper proposes to solve the inner problem of finding an adversarial perturbation by maximizing the margin gap of the classifier. Through experiments, the paper demonstrates that performing AT with the proposed attack leads to competent robust accuracies. Through experiments, it is also shown that the proposed change to AT also prevents robust overfitting phenomenon.

**Strengths:**

- The paper correctly identifies the problem of using improper surrogates for the inner maximization in AT.
- The paper hypothesizes that robust overfitting (RO) may be due to this problem of using improper surrogates for solving inner maximization in AT. Experimentally, it shows that RO can be eliminated by fixing this issue. This, I think, is surprising and worth exploring more from a theoretical viewpoint.
- The proposed modification to AT is simple, principled, and free of heuristics.

**Weaknesses:**

**Misleading presentation:**

I don't think the paper substantiates the claim made in the title, "AT should be cast as a non-zero sum game". The proposed "non-zero sum" formulation of AT in equations (16)-(17) is simply a tractactable reformulation of the standard zero-sum formulation in (11)-(12). The change is only in how the inner maximization is solved i.e., how the worst-case adversarial perturbation is found.

In Section 2.3 the paper claims that the tradeoff between robustness and accuracy as studied in many papers (in particular, Tsipras et. al. ICLR 2019) is a pitfall of the zero-sum formulation. However, the paper does not discuss how their supposed non-zero sum formulation resolves / ameliorates this trade-off either in theory or through experiments. There is also some misrepresentation of the prior work. For example, the robustness-accuracy tradeoff shown in Tsipras et. al. holds for the 0-1 robust loss. Clearly, the tradeoff is not caused by the use of improper surrogate loss.

Another claim is that the non-zero sum formulation eliminates robust overfitting. However, there is again no discussion on how the proposed non-zero sum formulation helps with this.

Overall, the paper repeatedly stresses on the non-zero sum property of their new AT formulation, and how it overcomes the pitfalls of the standard zero-sum formulation. But the main contribution of the paper is a tractable reformulation of the same old zero-sum game, by addressing the problems that occur through the use of surrogate losses.

There are other instances of misleading presentation in addition to the above. For instance, section 3.1 is titled "decoupling adversarial attacks and defenses". I don't think the proposed formulation (16)-(17) decouples the attacks and defenses any more than the standard formulation. The outer minimization still needs solution to the inner maximization.

**Missing prior work:**

While the main contributions of the paper (i.e. the reformulation of AT and the algo to solve the reformulated objective) stem from the core issue of using surrogate losses for AT, there is a surprising lack of any discussion on the many works that study surrogate losses for AT. See for instance Bao et. al. COLT 2020, Awasthi et. al. Neurips 2021 and the references therein.

Another surprising omission is any discussion on consistency. I quote from the paper, "Crucially, the inequality in (2) guarantees that the problem in (3) provides a solution that decreases the classification error (Bartlett et al., 2006), which, as discussed above, is the primary goal in supervised classification." This quote misses the main point of Bartlett et al., 2006. The solution to the surrogate loss minimization guarantees a solution to the 0-1 error minimization *if* the surrogate loss is "calibrated" or "consistent". While calibration and consistency are equivalent notions in standard classification (as shown in Bartlett et al., 2006), they are two separate things when it comes to adversarial classification. See the discussion in Munier et. al. Neurips 2022, and more recently Frank and Niles-Weed Neurips 2023.

**Questions:**

- I would really appreciate your comments on both the main weaknesses that I listed above.
- It would be good to include a toy example where the proposed inner maximization retrieves the optimal adversarial perturbation while the standard inner max with surrogate loss does not do so.
- How effective is BETA attack with a single PGD step, compared to FGSM?

---

> ### Author Response · Authors · 2023-11-21
> **Response Part 1: Clarifications regarding our presentation**
>
> Thank you for reviewing our work!  Here are some detailed comments which we believe resolve all of your main concerns.
>
> ---
>
> ## Presentation
>
> > "I don't think the paper substantiates the claim made in the title, "AT should be cast as a non-zero sum game". The proposed "non-zero sum" formulation of AT in equations (16)-(17) is simply a tractactable reformulation of the standard zero-sum formulation in (11)-(12). The change is only in how the inner maximization is solved i.e., how the worst-case adversarial perturbation is found."
>
> The short answer to your question is this: The validity of the title depends on what is meant by "adversarial training."  In much of the literature on *empirical* adversarial training (AT), AT is *defined* to be a zero-sum game w/r/t a *surrogate loss*. This is the case in several influential papers, e.g.,
>
> * Eq. (2.1) in the PGD paper [A]
> * Eq. (1) in the MART paper [B]
> * Eq. (1) in the FAST-AT paper [C]
> * Eq. (1) in the current top entry on the RobustBench leaderboard [D]
>
> One of the central messages of this paper is to guide the community away from this misconception; that is, we argue that the zero-sum formulation on the surrogate loss is *not* the correct starting point for AT.  And based on your reviwer, it sounds like we are in agreement about this, since indeed, if you accept the 0-1 loss zero-sum formulation, then our results could be more compactly summarized as "a tractable reformulation" of the 0-1 problem.  We would be happy to add the above discussion to our paper if you feel this would clarify our contribution.
>
> > "In Section 2.3 the paper claims that the tradeoff between robustness and accuracy as studied in many papers (in particular, Tsipras et. al. ICLR 2019) is a pitfall of the zero-sum formulation. However, the paper does not discuss how their supposed non-zero sum formulation resolves / ameliorates this trade-off either in theory or through experiments. . ."
>
> This is a fair point -- (Tsipras et al., 2019) considers the 0-1 zero-sum formulation, and thus there's reason to believe that the trade-off is not a pitfall of the zero-sum formulation.  Of course, there are many papers that observe such trade-offs in the surrogate zero-sum formulation, e.g., Section 5 in [E] and Section 4.3 in [F], so it could be the case that the surrogate makes the trade-off more pronounced.  But in any case, the goal of this part of the paper was to motivate why one might be interested in designing new ways to perform AT, and indeed we should have been more meticulous in designing our argument.
>
> Here's what we propose as a solution.  We will center our discussion of the pitfalls of AT on (a) *robust overfitting*, since (as far as we know) this phenomenon is specific to the surrogate-based zero-sum formulation, and (b) *the need for heuristics in SOTA attacks*, since this is widely observed in the literature regarding empirical robustness.  This also fits more tightly with our experiments, which focus on robust overfitting and test-time attacks.  Let us know what you think about this.
>
> > Another claim is that the non-zero sum formulation eliminates robust overfitting. However, there is again no discussion on how the proposed non-zero sum formulation helps with this.
>
> We provide this discussion in Section 5, paragraph 3 (entitled "BETA-AT outperforms baselines on the last iterate of training").  Based on your comment, our suspicion is that you are looking for a *theoretical* explanation, rather than an *empirical* demonstration (which was provided in the paper).  Frankly, we are also interested in proving (in some appropriate way) (a) why standard (i.e., surrogate) AT suffers from robust overfitting, and (b) why BETA-AT seems to resolve this, which together would constitute a more complete explanation.  However, we feel that our empirical evidence would already be of substantial interest to the ICLR research community, despite the fact that we have not completely answered the robust overfitting question in this paper.
>
> Full disclosure: A longer term research goal of ours is, having demonstrated empirically that BETA-AT seems to ameliorate robust overfitting, to attempt to prove why this phenomenon occurs.  However, we feel that this is a direction that should be left to future work, although we would certainly be open to discussing this further with you.

---

> ### Author Response · Authors · 2023-11-21
> **Response Part 2: Clarifications regarding prior work**
>
> > Overall, the paper repeatedly stresses on the non-zero sum property of their new AT formulation, and how it overcomes the pitfalls of the standard zero-sum formulation. But the main contribution of the paper is a tractable reformulation of the same old zero-sum game, by addressing the problems that occur through the use of surrogate losses.
>
> We think that our answer to the first comment in this response also applies here: The "non-zer-sum property" being utile *depends on the starting point*.  As before, if you think our paper would be stronger/clearer/more accurate if we were to place more emphasis on the "tractable reformulation" framing, we would be happy to do so.
>
> > There are other instances of misleading presentation in addition to the above. For instance, section 3.1 is titled "decoupling adversarial attacks and defenses". I don't think the proposed formulation (16)-(17) decouples the attacks and defenses any more than the standard formulation. The outer minimization still needs solution to the inner maximization.
>
> Perhaps the point of disagreement here is in the use of the term "decoupling."  Indeed, the defender still needs a solution to the inner problem, and so in some sense, the problems are still *coupled*.  Our choice of the word "decoupled" was meant to refer to the fact that the attack and defense players no longer optimize the same objective in (16)-(17), and therefore their objectives are not *coupled* in the same was as in the surrogate zero-sum version.
>
> We agree that some readers may find this confusing.  To resolve this, we would be happy to remove the word "decoupling" from our paper.  For instance, the last sentence before the start of Section 3.1, which originally read as follows:
>
> > ". . . we resolve this tension by decoupling the optimization problems of the adversary and the training algorithm."
>
> would be changed in the folloiwng way:
>
> > ". . . we resolve this tension by designing an optimization problem where the attack and defender optimize separate objectives."
>
> Let us know if this resolves your comment.
>
> ---
>
> ## Prior work
>
> > "While the main contributions of the paper (i.e. the reformulation of AT and the algo to solve the reformulated objective) stem from the core issue of using surrogate losses for AT, there is a surprising lack of any discussion on the many works that study surrogate losses for AT. See for instance Bao et. al. COLT 2020, Awasthi et. al. Neurips 2021 and the references therein."
> > "Another surprising omission is any discussion on consistency. . ."
>
>
> We agree, this literature should be discussed in more detail. We have added the following paragraphs to Section 6 of our updated PDF (in green):
>
> > In this paper, we focused on the *empirical* performance of adversarial training in the context of the literature concerning adversarial examples in computer vision.  However, the study of the efficacy of surrogate losses in minimizing the target 0-1 loss is a well studied topic among theorists.  Specifically, this literature considers two notions of minimizers for the surrogate loss also minimizing the target loss: (1) consistency, which requires uniform convergence, and (2) calibration, which requires the weaker notion of pointwise convergence (although [G] shows that these notions are equivalent for standard, i.e., non-adversarial, classification).
>
> > In the particular case of classification in the presence of adversaries, [H,J] claimed that for the class of linear models, no convex surrogate loss is calibrated w/r/t the 0-1 zero-sum formulation of AT, although certain classes of nonconvex losses can maintain calibration for such settings.  However, in [I], the authors challenge this claim, and generalize the calibration results considered by [H] beyond linear models.  One interesting direction future work would be to provide a theoretical analysis of BETA w/r/t the margin-based consistency results proved very recently in [L].  We also note that in parallel, efforts have been made to design algorithms that are approximately calibrated, leading to---among other things---the TRADES AT algorithm [K], which we compare to in Section 5.  Our work is in the same vein, although BETA does not require approximating a divergence term, which leads to non-calibration of the TRADES objective.`

---

> ### Author Response · Authors · 2023-11-21
> **Response Part 3: Answers to your questions**
>
> ## Questions
>
> > "It would be good to include a toy example where the proposed inner maximization retrieves the optimal adversarial perturbation while the standard inner max with surrogate loss does not do so."
>
> We have derived such an example during the rebuttal period.  Please see the newly added Appendix C in our paper.
>
> > "How effective is BETA attack with a single PGD step, compared to FGSM?"
>
> We are running this experiment now.  We are hoping to be able to give you an answer to this question before the review period ends.
>
>
>
> ---
>
> [A] Madry, Aleksander, et al. "Towards deep learning models resistant to adversarial attacks." arXiv preprint arXiv:1706.06083 (2017).
>
> [B] Wang, Yisen, et al. "Improving adversarial robustness requires revisiting misclassified examples." International conference on learning representations. 2019.
>
> [C] Wong, Eric, Leslie Rice, and J. Zico Kolter. "Fast is better than free: Revisiting adversarial training." arXiv preprint arXiv:2001.03994 (2020).
>
> [D] Peng, ShengYun, et al. "Robust principles: Architectural design principles for adversarially robust cnns." arXiv preprint arXiv:2308.16258 (2023).
>
> [E] Yang, Yao-Yuan, et al. "A closer look at accuracy vs. robustness." Advances in neural information processing systems 33 (2020): 8588-8601.
>
> [F] Raghunathan, Aditi, et al. "Understanding and mitigating the tradeoff between robustness and accuracy." arXiv preprint arXiv:2002.10716 (2020).
>
> [G] Bartlett, Peter L., Michael I. Jordan, and Jon D. McAuliffe. "Convexity, classification, and risk bounds." Journal of the American Statistical Association 101.473 (2006): 138-156.
>
> [H] Bao, Han, Clay Scott, and Masashi Sugiyama. "Calibrated surrogate losses for adversarially robust classification." Conference on Learning Theory. PMLR, 2020.
>
> [I] Awasthi, Pranjal, et al. "Calibration and consistency of adversarial surrogate losses." Advances in Neural Information Processing Systems 34 (2021): 9804-9815.
>
> [J] Meunier, Laurent, et al. "Towards Consistency in Adversarial Classification." Advances in Neural Information Processing Systems 35 (2022): 8538-8549.
>
> [K] Zhang, Hongyang, et al. "Theoretically principled trade-off between robustness and accuracy." International conference on machine learning. PMLR, 2019.
>
> [L] Frank, Natalie, and Jonathan Niles-Weed. "The Adversarial Consistency of Surrogate Risks for Binary Classification." arXiv preprint arXiv:2305.09956 (2023).

---

### Official Review · Reviewer_hcTf · 2023-11-02

**Soundness:** 3 good
**Presentation:** 3 good
**Contribution:** 3 good
**Rating:** 6
**Confidence:** 4

**Summary:**

The authors propose to use different surrogate loss for attacker and defender in adversarial training and formulate it as a non-zero-sum game and a bilevel optimization problem. A new adversarial training algorithm is proposed (BETA-AT), which is an extension of the TRADES method at high level. The authors claim that the algorithm eliminates robust overfitting and achieves state-of-the-art robustness. Numerical results on the CIFAR-10 dataset are reported.

**Strengths:**

The authors discuss why traditional adversarial training suffers from the zero-sum-game formula with a shared objective function in details. Also, the numerical results show that the proposed method achieves equal or better performance than some existing methods.

**Weaknesses:**

1. I feel the contribution of this work is somewhat limited in the sense that the core idea of the algorithm is not very new. At high level, the proposed BETA-AT method is an extension of TRADES: in TRADES training, the attacker aims to maximize $\max_{j\neq f_\theta(X)}M_\theta(X+\eta, f_\theta(X))_j$, and BETA gives a try with each incorrect class. I am wondering if it is the usual case that the BETA-AT gives the same attack as from TRADES, given the classifier is correct?

2. I expect more results for the experiments. Specifically, please consider the followings:

a) How does the training curve (as similar to Figure 1) of TRADES look like? Does it suffer robust overfitting in your setting?

b) Does BETA-AT take significantly longer time to train? Can you provide numerical results comparing the training time of the methods considered in Table 1?

**Questions:**

Please consider my questions in the previous section. I do not have additional questions in this section.

---

> ### Author Response · Authors · 2023-11-21
> **Clarifications regarding our contributions, and answers to your questions**
>
> Thank you for reviewing our work!  Here are some detailed comments which we believe resolve all of your main concerns.
>
> ---
>
> ## Our contribution
>
> > "I feel the contribution of this work is somewhat limited in the sense that the core idea of the algorithm is not very new."
>
> We disagree with this statement for two reasons.
>
> * **Lack of evidence.** We are not aware of any evidence that BETA-AT is "not very new."  Indeed, many AT schemes exist in the literature, but it's also the case that none of these algorithms are equivalent to BETA-AT, and crucially, none of them have the properties that we observed for BETA-AT (e.g., elimination of robust overfitting, lack of heuristics, etc.).  Therefore, we feel strongly that the "core idea" behind our algorithm is, in fact, "new."
> * **Our other contributions.** Of the four contributions we claim in the introduction, only one involves proposing BETA-AT.  Our other contributions include showing that BETA-AT eliminates robust overfitting and showing that BETA matches the performance of AutoAttack despite running 5.11 times faster.  We are absolutely sure that neither of these contributions has been shown before in prior work.  We'd appreciate your feedback regarding this point.
>
> We would be happy to discuss either of these points with you in more detail.
>
> ---
>
> ## Experiments
>
> > At high level, the proposed BETA-AT method is an extension of TRADES: in TRADES training, the attacker aims to maximize [the negative margin], and BETA gives a try with each incorrect class.
>
> This is incorrect for several reasons.  Consider the following:
> * **TRADES.** When running TRADES, one minimizes the clean risk plus a penalty on the KL divergence between the predictions for a clean data point and an adversarially perturbed copy.
> * **BETA-AT**  When running BETA-AT, one minimizes the risk of data points that have been adversarially perturbed by maximizing the negative margin.
>
> Therefore, not only do TRADES and BETA-AT optimize completely different objectives (TRADES uses a penalty whereas BETA-AT does not), these algorithms find adversarial examples in different ways.  That is, TRADES maximizes the KL divergence, whereas BETA-AT maximizes the negative margin; these quantites are *not* equivalent.  Please let us know if this explanation makes sense.  We would be happy to incorporate it into our paper if you think it would clarify our contribution.
>
> > I am wondering if it is the usual case that the BETA-AT gives the same attack as from TRADES, given the classifier is correct?"
>
> No, this is not the usual case.  Since TRADES and BETA-AT are fundamentally different algorithms (see the discussion to the previous comment), they do not constitute the "same attack."
>
> > "I expect more results for the experiments. Specifically, please consider the followings: (a) How does the training curve (as similar to Figure 1) of TRADES look like? Does it suffer robust overfitting in your setting? (b) (b) Does BETA-AT take significantly longer time to train? Can you provide numerical results comparing the training time of the methods considered in Table 1?"
>
> Regarding (a) -- we provided evidence in Table 1 that TRADES suffers from robust overfitting, just as PGD does.  We are curious: Did you find this evidence unconvincing regarding TRADES?  That TRADES displays robust overfitting is well known; see, e.g., Figure 3 in the TRADES paper, which does precisely the experiment you mention.  Of note is the fact that we used the authors' implementation of TRADES (https://github.com/yaodongyu/TRADES).
>
> Regarding (b) -- this plot was provided in Appendix B of our original submission; see Figure 3. We hope that this answers your question regarding the training time of the methods.  You'll notice in this plot that BETA-AT tends to take longer to run than PGD and TRADES, although as an evaluation attack, BETA is 5.11 times faster than AutoAttack without sacrificing any nominal performance.  That BETA-AT is computationally more expensive is a *design trade-off*: the improvements offered by BETA-AT (e.g., elimination of robust overfitting, the lack of heuristics, etc.) comes at a generally manageable cost of slightly longer training times.
>
> Please let us know if this resolves your concerns.
>
> ---
>
> [A] Rice, Leslie, Eric Wong, and Zico Kolter. "Overfitting in adversarially robust deep learning." International Conference on Machine Learning. PMLR, 2020.

---

> ### Comment · Reviewer_hcTf · 2023-11-22
> **Response to authors**
>
> Thanks for the clarifications and reminders. I do find most of my questions are addressed. I agree that the BETA-AT method is different from TRADES in the formulation of the objective. The penalty term considered by TRADES is closely related to the margin, while not equivalent.
>
> As for the experiments, I agree that the BETA-AT can avoid robust overfitting under the setting, with 3-4 times of computational cost compared to PGD and TRADES. The result on accuracy is significant. One follow-up question might be whether one can trade-off the accuracy with computational time based on BETA-AT, while maintaining the properly of anti-robust-overfitting.
>
> Considering the above, I would be happy to change my score to 6.

---

### Official Review · Reviewer_XrbP · 2023-11-04

**Soundness:** 3 good
**Presentation:** 3 good
**Contribution:** 2 fair
**Rating:** 6
**Confidence:** 4

**Summary:**

The paper attempts to show the standard zero-sum formulation of the adversarial training methods could lead to suboptimal results in practice. To resolve the issue, the authors propose a non-zero-game formulation of adversarial training problems where the classifier is trained using adversarial perturbations designed by optimizing a different margin-based loss function. This proposal results in the BETA-AT algorithm (Algorithm 2 in the paper). The numerical results in section 5 suggest improved results by applying the proposed BETA-AT method.

**Strengths:**

1- The paper discusses the potential drawbacks of the zero-sum game approach to adversarial training which I find interesting.

2- The paper numerically shows that the BETA-AT algorithm could improve the robustness of the neural nets and alleviate the robust overfitting issue.

**Weaknesses:**

1- While I find the paper's discussion on the drawbacks of zero-sum game formulation of adversarial training very interesting, I think the paper's proposal of considering a non-zero-sum game formulation for AT is not novel. The non-zero-sum game formulation of AT in Section 3.2 can be viewed as "applying an effective attack algorithm to generate adversarial examples for training the neural network". In that sense, the paper's main point could be viewed as "one should use effective attack algorithms to find adversarial examples in adversarial training". While this idea makes sense, some known variants of AT, e.g. adversarial training with examples generated by DeepFool attack, follow the same non-zero-sum game strategy: we do not optimize the adversarial perturbation by fully maximizing the classification loss and instead use a more effective attack algorithm.

2- Based on the point I discussed above, it seems that the paper's main contribution is to propose an adversarial attack scheme using the margin-based loss function (equation 13) and then use this attack scheme for adversarial training. While the proposed cost function makes sense, the paper does not demonstrate that the margin-based function possesses a desired property that sets it apart from other attack schemes. I think the paper would be much stronger if it theoretically showed what is unique about the specific margin-based attack scheme. Does this attack scheme help with the optimization or generalization of the AT method? At this point, it is not clear how the authors theoretically justify the choice of cost function (13) over other adversarial attack algorithms like DeepFool.

3- The paper's claim that the BETA-AT method helps with the robust overfitting issue is not well justified. The authors mention that "AT algorithms which seek to solve (8) are ineffective in that they do not optimize the worst-case classification error. For this reason, it should not be surprising that robust overfitting occurs." However, the overfitted AT models usually achieve 100% training accuracy under any norm-bounded adversarial attack so they perform perfectly against any norm-bounded perturbation to training data. Does not their perfect training performance show that they have been trained against strong enough adversarial attacks on training data?

I look forward to the authors' responses to the above questions and comments to assign my final score.

**Questions:**

1- What is the theoretical justification for applying the margin-based attack in (13) for designing adversarial examples over other adversarial attack methods? Does this attack scheme perform more effectively than other attack methods such as PGD or DeepFool? What is its fooling rate vs. the baseline attack methods when applied to standard image datasets?

2- (Weakness 3 above) The overfitted AT models usually achieve 100% training accuracy under any norm-bounded adversarial attack so they perform perfectly against any norm-bounded perturbation to training data. Does not their perfect training performance show that they have been trained against strong enough adversarial attacks on training data?

3- Figiure 1 shows the training and test accuracy of the networks until epock 130~140. Does the same generalization comparison remain valid if the training continues for more epcohs (like 200 epochs)?

---

> ### Author Response · Authors · 2023-11-21
> **Response Part 1: Clarifications regarding our contributions and properties of our formulation**
>
> Thank you for reviewing our work!  Here are some detailed comments which we believe resolve all of your main concerns.
>
> ---
>
> ## Our contributions
>
> > ". . . I think the paper's proposal of considering a non-zero-sum game formulation for AT is not novel.  The non-zero-sum game formulation of AT in Section 3.2 can be viewed as 'applying an effective attack algorithm to generate adversarial examples for training the neural network'.  In that sense, the paper's main point could be viewed as 'one should use effective attack algorithms to find adversarial examples in adversarial training'."
>
> We respectfully disagree, but we'd welcome a discussion on this topic.  Our main contribution is not just the fact that we view adversarial training as a non-zero-sum game; as you rightly mention, other works (including DeepFool) have studied the problem in this context.  Our main contribution is showing how this non-zero-sum perspective can be used to derive a *novel* algorithm---namely, BETA-AT---that provides robustness for DNN-based classifiers without commonly observed pitfalls (e.g., the need for heuristics and/or ensembles of distinct attacks).  We hope you agree that this framing constitues a clear contribution to the field.
>
> Based on our understanding of your review, it sounds like this criticism could be resolved by rephrasing parts of our paper, particularly w/r/t how we discuss our main contributions.  We have made some tweaks in our updated PDF in line with your suggestions, but if you have any suggestions, we would be happy to incorporate them.
>
> ---
>
> ## Properties of the non-zero-sum formulation
>
> > ". . . it seems that the paper's main contribution is to propose an adversarial attack scheme using the margin-based loss function (equation 13) and then use this attack scheme for adversarial training.
>
> We agree with this assessment.  If you think it would make our paper stronger, we would be happy to rephrase our contributions along these lines.
>
> > "While the proposed cost function makes sense, the paper does not demonstrate that the margin-based function possesses a desired property that sets it apart from other attack schemes. I think the paper would be much stronger if it theoretically showed what is unique about the specific margin-based attack scheme. Does this attack scheme help with the optimization or generalization of the AT method?"
>
> Thanks for this suggestion.  We agree that such a demonstration would make our paper stronger.  Therefore, in line with your comments, in the newly added Appendix C in our paper, we derived an example wherein our margin-based inner maximization retrieves the optimal adversarial perturbation while the standard inner max with surrogate loss fails to do so.  We hope you agree that incorporating this into our paper will improve our contribution.  Please let us know what you think.
>
> ---
>
> ## Robust overfitting
>
> > "The paper's claim that the BETA-AT method helps with the robust overfitting issue is not well justified."
>
> We respectfully disagree.  We provided strong evidence in Figure 1 and in Table 1 that BETA-AT mitigates robust overfitting.  This evidence shows that BETA-AT produces classifiers whose adversarial performance on the test set does not degrade after the first learning rate decay step.  We see this as clear justification for our claim, and if the reviewer disagrees, we would be open to hearing why you do not agree with the evidence we have provided.
>
> > "The authors mention that "AT algorithms which seek to solve (8) are ineffective in that they do not optimize the worst-case classification error. For this reason, it should not be surprising that robust overfitting occurs." However, the overfitted AT models usually achieve 100% training accuracy under any norm-bounded adversarial attack so they perform perfectly against any norm-bounded perturbation to training data."
>
> Your understanding here is correct -- when models are trained via AT, generally speaking, they overfit to the training data in the adversarial sense.  This is the case in, e.g., the results shown in Figure 1 of the original robust overfitting paper [A].
>
> > Does not their perfect training performance show that they have been trained against strong enough adversarial attacks on training data?"
>
> No, overfitting the training data is not sufficient to achieve robust generalization to test data which has been adversarially perturbed.  In other words, if a model has robustly overfit to the training data, it is often *not* the case that the model performs well when an adversary perturbs the test data.  This is one of the core messages of [A] (see, e.g., Figure 1 in [A]).  Please let us know whether this makes sense, and/or if we have understood your question correctly.

---

> ### Author Response · Authors · 2023-11-21
> **Response Part 2: Answers to your questions**
>
> ## Questions
>
> > "What is the theoretical justification for applying the margin-based attack in (13) for designing adversarial examples over other adversarial attack methods?"
>
> Please see the example we added at the bottom of the "Properties  of the non-zero-sum formulation" of this review.  We believe that this completely resolves your concern.
>
> > "Does this attack scheme perform more effectively than other attack methods such as PGD or DeepFool?"
>
> Yes -- this is the main message of Table 2 in our paper.  PGD and DeepFool are known to be weaker attacks than AutoAttack on standard image benchmarks; see, e.g., Table 1 in [B]. And in Table 2 of our paper, we show that BETA-AT performs almost identically to AutoAttack.  If this contribution is not clear in the current version of our paper, we are open to any suggestions regarding how we could highlight this aspect of our paper in a clearer way.
>
> > "The overfitted AT models usually achieve 100% training accuracy under any norm-bounded adversarial attack so they perform perfectly against any norm-bounded perturbation to training data. Does not their perfect training performance show that they have been trained against strong enough adversarial attacks on training data?"
>
> Please see the answer we gave at the end of the "Robust Overfitting" section of this response, which we believe answers this question.  If you would like us to elaborate, and especially if you feel that adding a clarification to our paper would strengthen our contribution, we would be more than happy to discuss this further with you.
>
> > "Figiure 1 shows the training and test accuracy of the networks until epock 130~140. Does the same generalization comparison remain valid if the training continues for more epcohs (like 200 epochs)?"
>
> Yes, it does.  We reran this experiment and observed almost identical results for BETA-AT, in that the test performance did not drop after the learning rate decay step.  We chose to cut Figure 1 off at 150 epochs on the x-axis because the plots are fairly uninteresting after that point, since BETA-AT just shows a flat line for the last 50 epochs.
>
>
> ---
>
> [A] Rice, Leslie, Eric Wong, and Zico Kolter. "Overfitting in adversarially robust deep learning." International Conference on Machine Learning. PMLR, 2020.
>
> [B] Croce, Francesco, and Matthias Hein. "Reliable evaluation of adversarial robustness with an ensemble of diverse parameter-free attacks." International conference on machine learning. PMLR, 2020.

---

### Official Review · Reviewer_DH4f · 2023-11-06

**Soundness:** 3 good
**Presentation:** 3 good
**Contribution:** 3 good
**Rating:** 6
**Confidence:** 3

**Summary:**

This paper proposed a robust training (AT) algorithm for deep neural networks. Different from most existing two-player zero-sum AT algorithms in which the attacker and defender simultaneously optimized the same objective, the proposed AT algorithm is formulated in a novel non-zero-sum fashion, where the attacker and defender optimized separate objectives. This separated scheme prevents the proposed AT algorithm from robust overfitting.

**Strengths:**

1. The proposed method resolves the issue of robust overfitting that is faced in existing zero-sum AT algorithms.
2. The paper is very well-written. The presentation is clear and easy to understand.
3. This paper points out the fundamental limitation in the popular zero-sum AT methods and provides a new non-zero-sum AT method, which would be an interesting future research direction for robust training.

**Weaknesses:**

1. The proposed AT algorithm has higher time complexity compared to zero-sum AT algorithms such as PGD, FGSM and TRADES.

**Questions:**

1. In Table 1. Why does BETA-AT$^{20}$ have lower test accuracy than BETA-AT$^{10}$ for clean data?

---

> ### Author Response · Authors · 2023-11-21
> **Clarifications regarding complexity, and answers to your questions**
>
> Thank you for reviewing our work!  Here are some detailed comments which we believe resolve all of your main concerns.
>
> ---
>
> ## Complexity
>
> > "The proposed AT algorithm has higher time complexity compared to zero-sum AT algorithms such as PGD, FGSM and TRADES."
>
> This is true.  In some sense, the benefits of BETA-AT (e.g., eliminating robust overfitting, matching the performance of AutoAttack without heuristics, etc.) all come at the cost of increased computation cost.  In Appendix B, we examine this tradeoff closely; we show that while is slower than PGD and TRADES, BETA is 5.11 times faster than AutoAttack.
>
> ---
>
> ## Questions
>
> > "In Table 1. Why does BETA-AT$^{20}$ have lower test accuracy than BETA-AT$^{10}$ for clean data?"
>
> This is the expected result and consistent with the vast majority of the literature in the field of adversarial robustness.  It is well known (see, e.g., [A,B,C]) that adversarial robustness is at odds with clean accuracy.  As an adversary gets stronger (i.e., it uses more steps of gradient ascent), the trade-off in clean accuracy becomes more pronounced.  In Table 1, as we use more steps of gradient ascent (i.e., we replace BETA-AT$^{10}$ with BETA-AT$^{20}$), the model becomes more robust, but clean accuracy falls.
>
> ---
>
> [A] Tsipras, Dimitris, et al. "Robustness may be at odds with accuracy." arXiv preprint arXiv:1805.12152 (2018).
>
> [B] Zhang, Hongyang, et al. "Theoretically principled trade-off between robustness and accuracy." International conference on machine learning. PMLR, 2019.
>
> [C] Dobriban, Edgar, et al. "Provable tradeoffs in adversarially robust classification." IEEE Transactions on Information Theory (2023).

---

> > ### Comment · Reviewer_DH4f · 2023-11-22
> >
> > I want to thank the authors for answering my questions and concerns. I would like to keep my score.

---

### Author Response · Authors · 2023-11-22
**Today (Wed., Nov. 22) is the last day of the discussion period**

Hi reviewers!

Today is the last day of the discussion period. We greatly appreciate your ongoing engagement with our paper and the feedback you have provided thus far.

* Reviewers `DH4f` and `hcTf` -- thank you for responding to our rebuttals!

* Reviewers `XrbP` and `NF3x` -- we would appreciate it if you could read through our rebuttals and let us know whether we have addressed your concerns.

Thanks,

The authors

---

### Meta-Review · Area_Chair_Q1vC · 2023-12-11

**Metareview:**

This paper makes an interesting argument to cast adversarial training as a non-zero-sum game to address the issue that surrogate losses are used instead of the 0-1 loss, and addressing problems with previous zero-sum surrogate formulations (in particular the so-called "robust overfitting" phenomenon).

After the rebuttal, all the reviewers recommended acceptance. The authors have made a strong rebuttal addressing most of the concerns from the reviewers, three reviewers updating their recommendation from 5 to 6 accordingly. It is important that the authors implement their promised changes in the camera ready version of the paper (in particular the ones mentioned for NF3x). As this was discussed with NF3x, I will note that I think the current title of the paper is appropriate (given that this is the formulation that they actually optimize, even if it is motivated from a 0-1 zero-sum formulation).

**Justification For Why Not Higher Score:**

This could also actually be a spotlight, depending on the SAC calibration. There is a vast literature on adversarial training, this paper provides an interestingly fresh and novel perspective on the formulation (with good empirical performance) that could interest a large public at ICLR.

**Justification For Why Not Lower Score:**

All reviewers recommend acceptance and this paper makes a very interesting contribution to ICLR.

---

### Decision · Program_Chairs · 2024-01-16

Accept (poster)